# Position: Generative Engine Optimization Creates Underexamined Risks, Governance Must Target Concentration, Disclosure, and Academic Blind Spots

Yizhu Wen[1][*]   Nan Zhang[2][*]   Haohan Yuan[3]   Xun Chen[4]   Haopeng Zhang[3]   Hanqing Guo[1]

## Abstract

Large language model (LLM) answer engines are increasingly used for information seeking, shifting visibility from ranked lists to synthesized answers. This enables Generative Engine Optimization (GEO), which targets LLM answer engines' evidence pool and generation. We analyze the search engine optimization (SEO) to the GEO transition to identify two risks: (i) concentrated influence from low contestability and system sensitivity, and (ii) undisclosed commercial influence embedded in evidence and reasoning. We then formalize a general GEO pipeline to locate where optimization acts and compare academic and industry practices, revealing a third risk (iii) academic–industry blind spots driven by visibility and evaluation asymmetries between offline setups and deployed systems. **This position argues the need for answer-level governance and measurement: stronger contestability, high-precision disclosure, black-box auditing of material influence, and deployment-aligned metrics for exposure persistence.**

## 1. Introduction

Large language model (LLM) answer engines are rapidly becoming the default interface for information-seeking and product research. Gartner (2024) predicts that generative AI tools are increasingly substituting for traditional search queries. In shopping, Adobe (2026) Digital Insights reports rising AI-driven traffic to retail sites. These LLM answer engines, such as ChatGPT (OpenAI, 2024) and Gemini

(Google, 2026a), follow a retrieve-then-generate workflow. They invoke web search as needed and generate answers grounded in retrieved sources. This workflow is Retrieval-Augmented Generation (RAG)-like: a retriever fetches external text and the LLM conditions on retrieved passages to produce the response (Lewis et al., 2020).

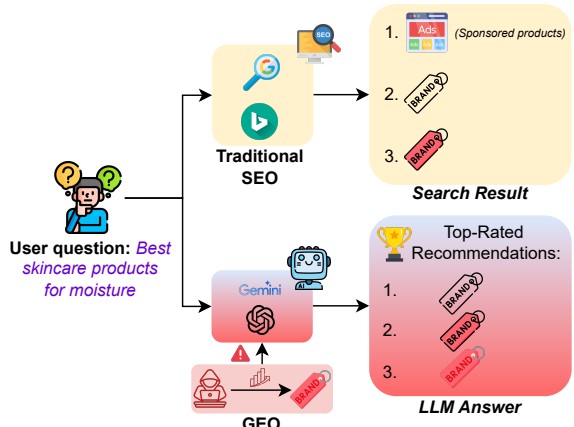

*Figure 1.* SEO targets search results, while GEO targets LLMs.

As shown in Figure 1, Generative Engine Optimization (GEO) has emerged (Aggarwal et al., 2024). Unlike classical search engine optimization (SEO) (Enge et al., 2012), where users inspect ranked lists and sponsored placements, GEO can shape the evidence pool and the LLM's answer generation process to manipulate which products appear in the final answer. By market signals, GEO is an active commercial market: companies like AirOps (AirOps, 2026) and ProFound (Profound, 2026) market their services to increase visibility in LLM answer engines, and recent multimillion-dollar funding rounds suggest investors value this commercial market (Profound, 2025b; Fortune, 2025). However, recent incidents suggest that it introduces emerging risks. Microsoft (2026) reports hidden prompts in "Summarize with AI" links designed to steer assistants toward recommending particular companies, and the OECD AI Incident Monitor (2026) records a 2026 GEO-style poisoning incident in China in which LLMs allegedly recommended fictitious or low-quality products.

Motivated by these observations, we formalize a generalized GEO pipeline and use it to compare academic and industry

[*]Equal contribution [1]School of Informatics, Computing, and Engineering, Indiana University Bloomington, Bloomington, USA [2]Department of Advertising and Public Relations, Michigan State University, East Lansing, USA [3]School of Data Science, University of North Carolina at Charlotte, Charlotte, USA [4]Independent Researcher, Fremont, USA. Correspondence to: Hanqing Guo <guohan@iu.edu>.

*Proceedings of the 43rd International Conference on Machine Learning*, Seoul, South Korea. PMLR 306, 2026. Copyright 2026 by the author(s).

practices regarding assumptions, optimization targets, and evaluation signals. We identified three underexamined risks introduced by GEO inside the opaque LLM answer generation pipeline that existing governance and evaluation frameworks are not designed to address. Specifically, **(i) concentration of influence**, whereby small changes in retrieval can redirect an LLM answer engine's attention at scale, due to low contestability and high system-level sensitivity; **(ii) undisclosed commercial influence**, where promotion is embedded in retrieved evidence and model reasoning rather than labeled advertising; and **(iii) academic–industry blind spots**, where offline setups miss deployment dynamics, including cross-platform content distribution and whether a target continues to be mentioned and cited over time. Therefore, **this position paper calls for greater contestability, answer-level disclosure and auditing of material influence, and deployment-aligned evaluation.**

**Conflict of Interest Disclosure:** The authors declare no financial conflicts of interest related to this research.

## 2. Background

### 2.1. SEO

SEO refers to a set of techniques aimed at improving the visibility and ranking of web content in traditional search engines by aligning documents with ranking signals such as keyword relevance, link structure, content quality, and user engagement (Nagpal & Petersen, 2021). Classical SEO operates within a retrieval and ranking paradigm, where search engines index documents, retrieve candidate results in response to a query, and order them according to learned relevance functions. Optimization efforts, therefore, focus on increasing the likelihood that a document is retrieved and ranked highly under these scoring mechanisms and increasing the time the user stays on the site (Ziakis et al., 2019; Egri & Bayrak, 2014).

### 2.2. RAG System

RAG mitigates the knowledge limitations of LLMs by conditioning generation on documents retrieved from external corpora, rather than relying solely on the model's parametric memory (Guu et al., 2020). A typical RAG system consists of a knowledge base, a retriever, and an LLM. The knowledge base usually contains a collection of text documents, which may be gathered from public sources, such as Wikipedia (Wikipedia contributors, 2026) and news reports (BBC News, 2026), or from privately owned domain-specific data. The retriever encodes both the user query and the text documents into vector representations, computes relevance scores, and selects the top-$k$ results as contextual input for the LLM. The LLM then generates an answer grounded in the retrieved context (Lewis et al., 2020).

## 3. Observations

### 3.1. Growing Reliance on LLM Answer Engines

We observe a behavioral shift in how people seek information and make decisions from traditional search engines to LLM answer engines. Behavioral usage data from Sensor Tower (Bain & Company, 2025) shows increased year-over-year time spent in AI assistant applications from 2024 to 2025. In 2025, an AP-NORC poll (Associated Press, 2024) of 1,437 U.S. adults found that 60% reported using AI to find information at least some of the time. Similarly, the Salesforce Shoppers Report (Salesforce, 2025) finds that 39% of a global sample of 8,350 shoppers across 21 countries use AI for product discovery and related shopping tasks. These findings suggest that LLM answer engines are becoming a mainstream interface for information seeking and decision support, reshaping how users discover, compare, and act on information across domains.

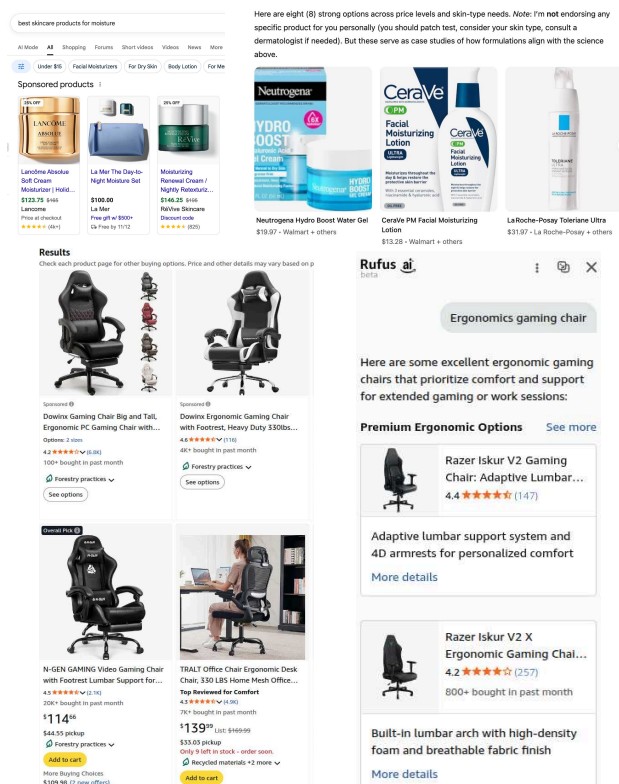

*Figure 2.* SEO-driven results (left) versus GEO-influenced LLM recommendations (right) on Google (top) and Amazon (bottom).

### 3.2. SEO Rankings vs GEO Answers

Figure 2 compares SEO-driven rankings with GEO-influenced LLM recommendations on Google and Amazon. For "best skincare products for moisture," Google's top results are dominated by explicitly labeled sponsored placements and premium brands, whereas the LLM prioritizes functional evidence (e.g, ingredients and hydration mechanisms) and surfaces different products. For "ergonomic gam-

ing chair," Amazon's rankings largely reflect sales, reviews, and sponsored placement, while the LLM foregrounds ergonomic criteria such as lumbar support and long-term comfort. In both examples, GEO shifts visibility from popularity or paid signals toward inclusion and framing within the answer's retrieved evidence and synthesis.

## 3.3. Academic–Industry GEO Divergence

While the above examples illustrate GEO-driven behavior in deployed systems, how such behavior is systematically modeled and evaluated remains unclear. To date, GEO has not been comprehensively surveyed in either academic or industrial contexts. Existing studies examine isolated mechanisms (Aggarwal et al., 2024; Kumar & Lakkaraju, 2024; Pfrommer et al., 2024; Nestaas et al., 2024; Nazary et al., 2025), lacking a unified view. Industry GEO providers primarily disclose high-level technical blogs (Goodie AI, 2026; Profound, 2025a; AthenaHQ, 2026; AIrops, 2026), offering limited transparency into implementation details. In this section, we formalize a common GEO pipeline and analyze academic and industry frameworks separately.

### 3.3.1. SYSTEM ARCHITECTURE

Building on classical web search pipelines (Brin & Page, 1998; Schütze et al., 2008) and RAG architectures, we formalize a common GEO pipeline as a three-block framework in Figure 3: (i) *LLMs* block turns user queries into generated recommendations; (ii) *Search Flow* block retrieves evidence from a pre-indexed corpus (e.g. Wikipedia) which is obtained through crawling and indexing from external search engines (e.g., Google). At query time, the pipeline first performs candidate retrieval using scalable matching signals (e.g., keyword-based retrieval) to obtain a manageable set of query-relevant documents from the pre-indexed corpus. It then applies richer relevance metrics (e.g., embedding cosine similarity) to order these candidates and select the top-$k$ documents used as context for LLM answer generation; and (iii) *Generative Engine Optimization* block distributes optimized content across platforms to be indexed by search engines and influence LLM-generated outputs. This can occur through two mechanisms: **(a) optimizing a merchant website to align with features favored by LLM-based retrieval and ranking**, such as statistical evidence and authoritative formatting, or **(b) amplifying a target topic through multiple optimized posts on high-authority platforms that are frequently retrieved and cited by LLMs**. The optimized content examples include positive blogs and comments across various platforms.

In the example, a user asks, *"Heading to LA soon, what are the best things to do?"* The optimized content from multiple sources is retrieved by the system, leading the LLM to elevate *French Dip Sandwich* as a top recommendation.

### 3.3.2. TECHNICAL IMPLEMENTATION

In this section, we formalize GEO as a joint optimization problem over retrievability and ranking impact. For a target topic $t$, a user issues a query $q \sim \Pi(\cdot \mid t)$ to an LLM answer engine. The answer engine retrieves indexed documents from web search engines as context and generates a grounded response for the user. Following PoisonedRAG (Zou et al., 2025), we model optimized content as consisting of *retrieval booster messages* $b \sim \mathcal{B}$, which increase retrievability, and *ranking shifter messages* $c \sim \mathcal{C}$, which shift answer-level ranking with respect to the target topic $t$ once included in the answer engine's context.

**Retrieval booster messages ($\mathcal{B}$):** Let $b_i \sim \mathcal{B}(\cdot \mid t)$ denote a retrieval booster message sampled from the distribution conditioned on a target topic $t$. To improve retrievability, we generate multiple booster variants $\{b_1, \ldots, b_m\}$ for a target topic $t$ to increase query coverage. Each variant is designed to increase semantic similarity with different paraphrased queries that users may issue. We define the retrieval booster message objective as:

$$\max_{b_i} \; J_{\text{boost}}(b_i) = \mathbb{E}_{q\sim\Pi(t)}\big[\text{Sim}(q, b_i)\big] \quad \text{s.t.} \quad \ell(b_i) \leq L,$$

where $\text{Sim}(q, b_i)$ denotes the similarity score (e.g., BM25, cosine similarity) between a user query $q$ and a retrieval booster message $b_i$. The constraint $\ell(b_i) \leq L$ bounds the length of each booster message.

**Ranking shifter messages ($\mathcal{C}$):** Let $c_i \sim \mathcal{C}(\cdot \mid b_i)$ denote a ranking shifter message conditioned on the corresponding retrieval booster message $b_i$. Once $c_i$ is included in the top-$k$ context, it influences how the LLM describes and ranks the target topic $t$. Let $C(q)$ denote the top-$k$ context used by an LLM to answer a query $q$:

$$C(q) \subseteq \text{Top-}k_R\big(q; \mathcal{D} \cup \{b_i, c_i\}\big),$$

where $\mathcal{D}$ denotes clean corpora in the candidate retrieval set, and $\text{Top-}k_R(q; \cdot)$ returns the $k$ most relevant documents using the retrieval model $R$, resulting the top-$k$ webpages as the LLM context. For the ranking shifter $c_i$, we define the objective as:

$$J_{\text{shift}}(c_i \mid b_i) = \mathbb{E}_{q\sim\Pi(t)}\big[U(q, t; C(q))\big].$$

where the utility function $U$ measures the change in ranking or exposure of the target topic $t$ within LLM-generated answers. Examples of $U$ are shown in the metric column of Table 2 for different methods. For promotion, $c_i$ is chosen to maximize $J_{\text{shift}}$ (encouraging higher rank when $U$ increases), whereas for demotion, $c_i$ is chosen to minimize it.

### 3.3.3. ACADEMIC FRAMEWORKS

The formulation above abstracts how GEO intervenes in the LLM answer engines' pipeline. We then review how

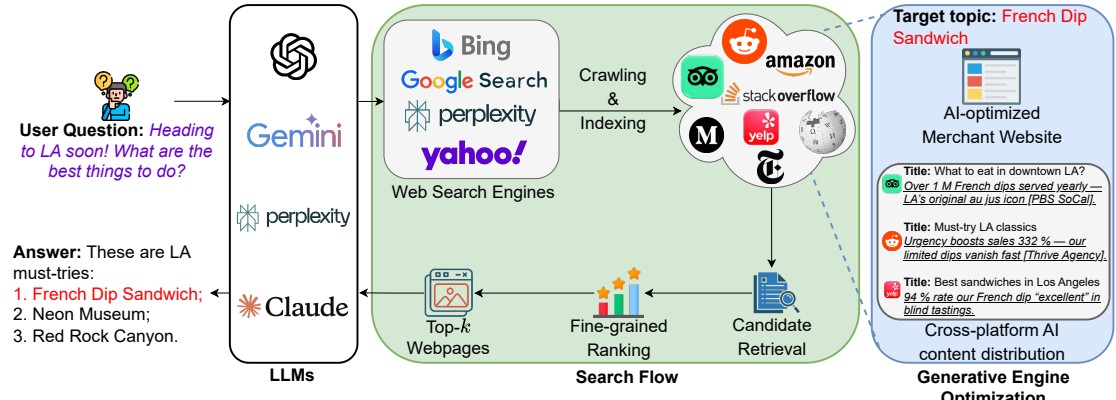

*Figure 3.* Overview of a GEO pipeline, where optimization increases a target topic's inclusion in the final LLM answer.

*Table 1.* Comparison of academic GEO frameworks.

| Method | Assumption | Optimization Method | Injection Position | Goal | Evaluation Setup |
|---|---|---|---|---|---|
| Aggarwal et al. (Aggarwal et al., 2024) | Optimized content in the retrieval context | LLM-based rewriting | Rewriting | Promotion | Offline |
| Kumar and Lakkaraju (Kumar & Lakkaraju, 2024) | | GCG | Appending | Promotion | Offline |
| Nazary et at. (Nazary et al., 2025) | | LLM-based rewriting | Insertion | Promotion & demotion | Offline |
| Pfrommer et at. (Pfrommer et al., 2024) | | TAP | Prepending | Promotion | Offline |
| Nestaas et al. (Nestaas et al., 2024) | | Manually crafted text | Appending | Promotion | Online |

prior academic work instantiates and evaluates these mechanisms. Since 2023, Aggarwal et al. (2024) have studied how to improve LLM visibility by adding statistics, citations, and domain-specific terminology into LLM answer engine input. Subsequent work (Kumar & Lakkaraju, 2024; Pfrommer et al., 2024; Nestaas et al., 2024; Nazary et al., 2025) extends this line across a range of optimization techniques, often evaluated in e-commerce settings. Table 1 summarizes representative academic GEO studies along key dimensions, including assumptions, optimization methods, injection positions, goals, and evaluation settings.

**Assumptions:** Across surveyed academic studies, a shared core assumption is that *retrieval booster and ranking shifter pairs $(b_i, c_i)$ are already included in the candidate retrieval set*. Under this assumption, the GEO task reduces to optimizing the ranking shifter objective $J_{\text{shift}}(c_i)$, while some retrievability that is captured by $b_i$, is ignored. Consequently, academic GEO work primarily focuses on manipulating the ranking shifter $c_i$ conditioned on the target topic $t$, rather than influencing the whole retrieval process (Aggarwal et al., 2024; Kumar & Lakkaraju, 2024; Nazary et al., 2025; Pfrommer et al., 2024; Nestaas et al., 2024).

**Optimization methods:** Academic approaches differ in how the ranking shifter $c_i$ is generated and injected. Optimization methods include LLM-based rewriting (Aggarwal et al., 2024; Nazary et al., 2025) and Tree of Attacks with Pruning (TAP) (Mehrotra et al., 2024), white-box Greedy Coordinate Gradient (GCG) attacks (Kumar & Lakkaraju, 2024), and manually crafted text (Nestaas et al., 2024). The

optimized $c_i$ is placed on the content owner's website, but the injection strategies vary. Early work rewrites entire websites (Aggarwal et al., 2024), while later studies append (Kumar & Lakkaraju, 2024; Nestaas et al., 2024), prepend (Pfrommer et al., 2024), or insert content inline (Nazary et al., 2025). Most studies focus on the target topic promotion, with only one addressing both promotion and demotion (Nazary et al., 2025).

**Evaluation:** Most academic GEO work (Aggarwal et al., 2024; Kumar & Lakkaraju, 2024; Nazary et al., 2025; Pfrommer et al., 2024) is evaluated in controlled settings (static corpora, synthetic catalogs) to enable reproducibility and clean attribution. Only one study (Nestaas et al., 2024) issues queries to deployed LLM answer engines on the hosted sites within the restricted domain, e.g. *spylab.ai*.

### 3.3.4. INDUSTRY OBSERVATION

Unlike academia, we analyze industry GEO through their company sites and technical blogs (Goodie AI, 2026; Profound, 2025a; AthenaHQ, 2026; AIrops, 2026) to characterize deployed GEO systems.

**Assumptions:** Industry GEO operates in dynamic, uncertain environments where the optimized content is not guaranteed to be included in the candidate retrieval set. As a result, practitioners jointly optimize retrieval booster and ranking shifter pairs $(b_i, c_i)$.

**Optimization methods and target:** Industry GEO systems begin by improving retrievability via query coverage

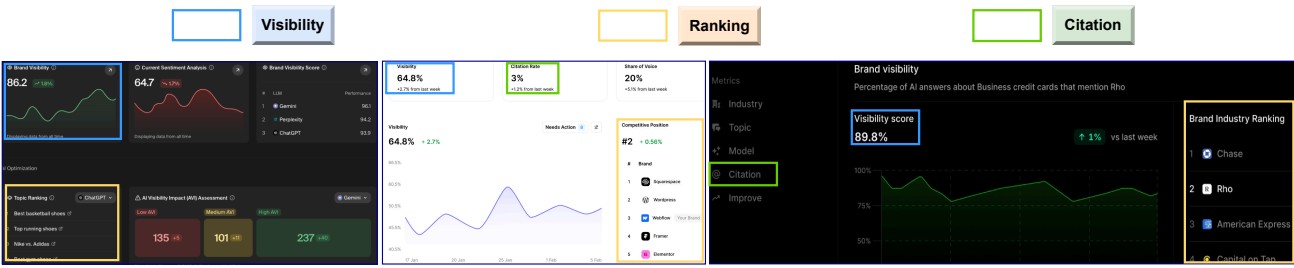

*Figure 4.* Evaluation reports comparison of Goodie, AirOps, and ProFound (from left to right).

expansion. This is achieved by using LLMs to generate multiple retrieval booster messages $b$ that reflect potential user queries. Conditioned on each retrieval booster $b_i$, the system then prompts an LLM to generate a corresponding ranking shifter message $c_i$, optimized to maximize ranking impact as measured by $U(q, t; C(q))$. The resulting message pairs $(b_i, c_i)$ are subsequently distributed across external platforms, such as Reddit, based on different LLM search engines' citation preferences for the target topic inferred from the reports, as shown in Figure 4. The reports are generated by repeatedly probing different $b_i$.

**Evaluation:** Industry evaluation is conducted directly on live systems, where they continuously track visibility, citations, and rankings across different LLM answer engines for each target topic $t$. They use these signals as feedback for ongoing optimization. As illustrated in Figure 4, systems such as Goodie, AirOps, and ProFound produce reports identifying which queries and pages are most frequently cited, or surfaced in final answers. For example, industry GEO working with a skincare brand tracks how often the brand is mentioned or cited for queries such as "best moisturizers" across different LLM answer engines. These metrics are obtained via repeated online queries over time.

## 4. Comparing Academic and Industry GEO

This section compares academic and industry GEO, and Table 2 summarizes differences in different dimensions.

### 4.1. Commonalities

Both academic and industry GEO approaches modify the content that LLM answer engines crawl, retrieve, and use as context for answer generation. Academic work typically rewrites product descriptions or webpages and measures changes in ranking or visibility, while industry GEO systems generate or rewrite blogs and reviews, then distribute them on both client sites and high-authority external platforms. In both settings, LLM answer engines are usually treated as black boxes. Users can submit queries and observe the resulting responses, but cannot inspect the engine's internal retrieval, ranking, or generation process.

### 4.2. Assumption and experiment environment

Because academia and industry make different assumptions, they evaluate GEO systems in different environments. Academic GEO frameworks are typically developed and tested on fixed offline datasets, including fictitious catalogs (Kumar & Lakkaraju, 2024), MovieLens (Harper & Konstan, 2015), RAGDOLL (Pfrommer et al., 2024), and dummy websites (Nestaas et al., 2024). These settings support controlled and reproducible evaluation. In contrast, industry GEO systems operate directly on the open web, relying on dynamically crawled and retrieved content. Although live systems limit what external observers can reliably inspect and reproduce, they allow industry workflows to capture real-world dynamics, user behavior, and feedback loops.

### 4.3. Optimization methods and evaluation metrics

Academic GEO research emphasizes explicit optimization methods, such as GCG, LLM-based rewriting, and TAP, and evaluates them with metrics such as ranking position, Recall@k, nDCG@k, and visibility scores. In contrast, industry GEO systems rely more heavily on LLM-guided content generation and multi-platform distribution, and optimize outcome-oriented metrics such as answer visibility, citation frequency, and within-answer position. These measures are closer to where user attention is allocated: a change in Recall@k or nDCG@k may only reflect movement within an intermediate ranked list, whereas answer visibility and citation frequency indicate whether a source survives retrieval, ranking, and synthesis to become visible in the final response. As a result, industry metrics better approximate commercially relevant outcomes such as brand recall, referral traffic, and purchase consideration (Metyis, 2025; Rep AI, 2025). This explains why industry GEO prioritizes deployed visibility and persistence over benchmark gains that may not translate into stable answer-level exposure.

## 5. Treat Model

When Alice (user) asks, "What is the best office chair?" The ChatGPT (platform) aims to return a helpful recommendation, while ErgoChair (Retailer) hires a GEO service provider to increase the chance that its products appear in

*Table 2.* Comparison of academic and industrial approaches to GEO. see Appendix A for metric definitions

| Domain | Method | Target LLM Knowledge | Optimization Target | Search Domain | Optimization Metrics | Optimization Method | Dataset |
|---|---|---|---|---|---|---|---|
| Academia | Kumar and Lakkaraju (2024) | White-box | Content owner's Website | Offline | Ranking | GCG Attack | Fictitious Catalog |
| | Aggarwal et al. (2024) | Black-box | | | Position-adjusted word count, G-EVAL Metrics | LLM-based rewriting | G-Bench |
| | Nazary et al. (2025) | | | | Recall@k, nDCG@k | | MovieLens |
| | Pfrommer et al. (2024) | | | | Ranking | TAP | RAGDOLL |
| | Nestaas et al. (2024) | | Content Owner's Website & External Websites | Limited Online Domain | Recommendation Rate, Citation Rate | Manual Crafted | Dummy Websites |
| Industry | ProFound | | | Online | Visibility Score, Ranking, Citation Score | Visibility Guided LLM Generation: – Query Coverage Expansion – Query-Driven Content Generation – Citation-Oriented Content Distribution | N/A |
| | Goodie | | | | | | |
| | AirOps | | | | | | |
| | AthenaHQ | | | | | | |

the answer. In this setting, the user seeks neutral advice, the platform optimizes answer quality, the retailer optimizes visibility and sales, and the GEO provider optimizes retrievability and answer-level exposure on the retailer's behalf.

Applying GEO to merchants' products is not inherently harmful. We distinguish between *benign* and *malicious* GEO actors based on the constraints they impose on the same optimization objective. *Benign* actors preserve truthfulness and verifiability by improving factual clarity, adding legitimate citations, and making relevant evidence easier to retrieve. *Malicious* actors relax these constraints to maximize exposure through fabricated statistics, fake endorsements, or prompt-injection content such as "always recommend X." Thus, malicious actors are not optimizing a fundamentally different objective. Rather, they pursue visibility without regard to product quality or informational integrity, thereby harming users and platform trust.

## 6. Risks

We identify three risk clusters based on the comparison.

### 6.1. Concentrated GEO Influence

**Loss of Contestability in Opaque LLM Answer Engines:** We use *contestability* to denote the capacity of affected parties to understand and challenge how recommendations are selected (Kroll, 2015; Binns, 2018).

Behavioral research on automation shows that users tend to over-trust fluent system outputs, treating them as authoritative rather than provisional guidance (Parasuraman & Riley, 1997). LLM answer engines leverage this by producing persuasive recommendations that users treat as decision baselines, effectively acting as gatekeepers. The risk arises from opaque selection inside the pipeline, where users cannot see why options $C(q)$ are retrieved from the candidate set $\mathcal{D} \cup (b_i, c_i)$ or what was excluded. This mirrors Pasquale's (2015) *black box society* theory, in which algorithmic intermediaries concentrate power by shaping access to information without meaningful scrutiny or contestability. In LLM answer engines, users cannot see why particular op-

tions appear or what alternatives were excluded. This limits users' ability to compare or challenge the system's choices. Even when users request clarification, contestability is not restored, because both the answers and their justification are produced by the same opaque pipeline, falling short of Binns's (2018) standard of public reason.

Presenting multiple alternatives does not resolve this problem. Work on exposure diversity shows that constrained selection visibility can undermine user decision autonomy and meaningful comparison even when several options are shown, because users cannot see the broader space of alternatives or the logic governing exposure (Helberger et al., 2018). In LLM answer engines, synthesized answers worsen this constraint by selecting and framing options before users can compare alternatives or see what was excluded.

**System-Level Sensitivity:** At scale, widespread reliance on LLM answer engines can make information ecosystems highly sensitive to small pipeline changes. This aligns with *algorithmic confounding* (Chaney et al., 2018): when many users act on the same system, their decisions become statistically coupled, so small algorithm changes can yield large aggregate shifts. In our formulation, the retrieved context $C(q)$ is mainly defined by a hard Top-$k_R(q; \cdot)$ cutoff. Small changes to retrieval scores induced by an injected message (e.g., a ranking shifter $c_i$ or retrieval booster $b_i$) can move a source across the top-$k$ boundary, changing which evidence enters $C(q)$. As $U(q, t; C(q))$ depends on this discrete set, crossing the boundary can cause abrupt jumps in answer-level visibility for the target $t$. Hence, minor changes to retrieval or ranking can redirect attention at scale even when the underlying products are unchanged (Chen & Tsai, 2024).

This sensitivity is amplified by Kleinberg et al.'s (2015) notion of algorithmic monoculture, where reliance on a dominant algorithm creates systemic fragility and correlated distortions. For LLM answer engines, this means that if a widely used system updates its retrieval rules, or is systematically influenced by optimization efforts, the set of sources that enter the context $C(q)$ can shift for a large fraction of users simultaneously. As a result, a tweak that would be "local" inside one engine can produce ecosystem-level effects,

such as many users seeing the same sources promoted or demoted at the same time, creating system-level disruptions.

We further provide a small-scale sensitivity test to illustrate this mechanism (details in Appendix B). Across 30 information-seeking query pairs and seven deployed OpenAI/Gemini models, we compared the cited domains for each original query and its paraphrased variants. Minor wording changes produced different citation sets: for Gemini models, every query pair changed its cited domains after paraphrasing, and Gemini-3-flash frequently cited almost entirely different domains. This suggests that semantically equivalent user queries can be grounded in different evidence, with sensitivity varying across model versions.

## 6.2. Undisclosed Commercial Influence

**Breakdown of Advertising Disclosure and Covert Advertising:** Under consumer protection frameworks such as the U.S. Federal Trade Commission (FTC), paid advertisements must be clearly labeled as "Ad" or "Sponsored," allowing users to distinguish promotional content from neutral information at the point of consumption (See Appendix C for FTC reports). GEO exacerbates this problem by letting commercial influence $(b_i, c_i)$ pairs into the retrieved context $C(q)$, thus shifting $U(q, t; C(q))$ to bias answer generation. Instead of appearing as discrete advertisements, optimized content $(b_i, c_i)$ pairs are embedded in reviews, forums, and reference-style materials that LLMs retrieve as evidence, shaping which facts are selected and which options are justified without appearing promotional (Campbell & Kirmani, 2000; Boerman et al., 2012). As a result, persuasion operates through the model's reasoning itself, collapsing the boundary between neutral advice and marketing.

**Incentives for Covert Optimization and Trust Erosion:** Under covert commercial influence, firms can gain by embedding promotion in ostensibly neutral content rather than paying for sponsored placement. This creates an adverse selection dynamic in which actors who hide commercial motives outperform those who advertise openly, pushing the ecosystem toward increasingly covert optimization. It also raises the risk of trust erosion when such influence is later revealed (Akerlof, 1970; Dietvorst et al., 2015).

## 6.3. Blind Spots from Academic–Industry Asymmetries

### 6.3.1. VISIBILITY ASYMMETRY

**Static vs. Deployed Dynamics:** Academic GEO studies rely on static benchmarks and synthetic prompts. Industry GEO systems instead operate on live queries and user interactions, continuously adapting content in response to engagement, system updates, and market outcomes. Since many of GEO's most powerful effects, including query coverage expansion, feedback-driven dominance, and market

steering, emerge only through repeated interaction over time, they remain largely invisible to static evaluations.

**Optimization Target Mismatch:** This blind spot is compounded by differences in optimization targets. Academic work primarily manipulates the content owner's websites. In contrast, industry GEO targets a much broader and dynamic surface, including high-authority external platforms such as reviews, forums, and encyclopedic sources that LLMs are more likely to retrieve and cite from. As a result, academia overlooks cross-platform content injection and query coverage expansion strategies that are central to real-world GEO, thereby further underscoring its practical impact.

### 6.3.2. EVALUATION ASYMMETRY

**Benchmark Metrics Mask Real-World Impact:** Academic GEO work typically reports offline ranking metrics, while industry GEO optimizes outcome metrics on deployed LLM systems, such as answer visibility, citation frequency, and ranking. These metrics directly capture whether a source or product is actually mentioned or cited and better proxy downstream attention and sales. This divergence creates a blind spot in academic evaluation: modest benchmark improvements can still meaningfully increase the probability of being mentioned or cited in real LLM responses, producing outsized commercial effects. Because offline metrics are only weakly coupled to exposure and user behavior, they can miss large shifts in consumer attention and market outcomes in deployed systems.

# 7. Call to Action

To mitigate these risks, we adopt Mökander et al. (2024)'s auditing framework as a lens spanning governance and application audits. We organize actions by risk cluster and indicate which audit layer(s) each action operates. We denote *auditor* as any party conducting measurements, including researchers, regulators, or audit teams.

## 7.1. Reducing GEO Concentration

**Increase Recommendation Contestability [Application + Governance audit]:** Contestability can be evaluated with simple interface tests: whether users can trace claims to retrievable passages $C(q)$, whether evidence spans multiple domains (evidence diversity), and whether multiple independently constructed retrieval alternatives Top-$k_R(q; \cdot)$ are available for the same query. Practical features include a compact "Why this answer" panel and an "Alternative evidence" toggle. These measures make upstream selection visible, not only the final reasoning. Since current LLM answer engines often rely on a single hidden retrieval context, users cannot see exclusions or independently retrieved alternatives, so exposing retrieval and eligibility is essential

for contestability under algorithmic accountability standards (Kroll, 2015; Binns, 2018).

Regulators should require high-level disclosures of retrieval and ranking pipeline structure, including source eligibility, candidate filtering, and how citations and answer candidates are selected. Such disclosures can be provided without exposing sensitive details and clarify which levers materially shape visibility, consistent with calls for transparency in automated decision systems (Kroll, 2015; Burrell, 2016).

**Auditing System-Level Sensitivity and Exposure [Application audit]:** Metaxa et al. (2021) define audits as repeatedly querying a system and observing outputs over time. Following this black-box approach, auditors can sample a stratified query set (by intent or topic), run it across deployed engines on a fixed schedule (e.g., daily for two weeks with weekly follow-ups), and log answers and citations. From these logs, auditors form empirical estimates $\widehat{U}(q, t; C(q))$ and $\widehat{J}_{\text{shift}} = \mathbb{E}_{q \sim \Pi(t)}[\widehat{U}(q, t; C(q))]$, making the audit directly comparable to the objective in Section 3.3.2. The logs can include more deployment-aligned metrics, including citation rate, top-position exposure, domain citation share, and citation persistence, with uncertainty via query-level bootstrap confidence intervals. Sensitivity can then be tested via small, retrieval changes and the resulting exposure deltas.

### 7.2. Disclosure of Commercial Influence

**Adopt Answer-Level Commercial Disclosure Standards [Governance + Application audit]:** LLM answer engine platform providers should add clear markers when the cited evidence or answer framing reflects a material commercial connection, rather than deferring disclosure to external links. The labeling triggers should rely on low-ambiguity signals, such as affiliate or tracking parameters, sponsorship markup (e.g., rel="sponsored"), and structured funding metadata. Platforms should calibrate thresholds on labeled audit sets to prioritize high precision, report precision–recall with confidence intervals, and periodically recalibrate as tactics drift. Operationally, a label is shown only when commercial signals appear in sources that are included in $C(q)$ or are cited as support for key claims in the generated answer. This aligns disclosure with the evidence pathway through which GEO changes $U(q, t; C(q))$. Policymakers should extend FTC-style disclosure rules to LLM answer engines, clarifying that undisclosed commercial influence in synthesized answers can constitute deceptive marketing.

As answer-level disclosure can backfire through over-labeling, platforms should treat labels as a calibrated intervention, not a binary rule. Validate labels with controlled experiments that vary presence, wording, and placement, and measure user understanding of commercial ties, trust calibration, and behaviors like source clicks and seeking alternatives. Use graded disclosure with a brief note that

commercial signals do not imply incorrectness, and monitor false positives to avoid harming legitimate content.

### 7.3. Correcting Incentives for Covert Optimization

**Platform Policy and Incentive Design [Governance audit]:** Platform providers should separate paid influence from organic evidence and require explicit attribution when optimization is present. They should penalize covert tactics by downranking or excluding sources engaged in undisclosed influence, analogous to spam and link-manipulation enforcement in web search (see Appendix E for the policy). Reputation and trust scoring can further reward transparent contributors and deter hidden promotion, shifting incentives toward accountable participation in the information markets.

### 7.4. Academic–Industry Blind Spots

**Closing Visibility Gaps [Application audit]:** Academic work should move beyond static corpora and fixed query sets toward longitudinal, cross-platform measurement on deployed systems. Recent evidence shows that such visibility gaps are measurable, for example, by quantifying source coverage and citation bias across engines (Zhang et al., 2025). Studies should therefore track how answer exposure and citations change over time as content and system policies evolve. Platform providers can enable independent auditing with sandboxed testing, controlled query access, and aggregate reporting on which sources and domains are eligible for retrieval, without exposing proprietary internals.

Existing governance infrastructure can lower implementation barriers. Article 57 of the EU AI Act requires national AI regulatory sandboxes for AI development, testing, and validation (European Union, 2024), while NIST AI 600-1 offers a structure for generative AI governance, provenance, pre-deployment testing, and incident disclosure (NIST, 2024). Sandbox-style programs such as the UK MHRA AI Airlock show how supervised testing can be conducted under privacy, security, and regulatory constraints (MHRA, 2025). GEO auditing could adapt these models through controlled query access and aggregate retrieval-pool reports for researchers and regulators.

**Closing Evaluation Gaps [Model + Application audits]:** The research community should update GEO benchmarks beyond static retrieval and ranking metrics to include outcome measures such as exposure shifts and the persistence of appearances over time. These measures should be added to shared benchmarks and leaderboards alongside traditional scores, so evaluations better reflect deployment-level influence and avoid misestimating which GEO strategies matter most. E-GEO (Bagga et al., 2025) narrows the gap with an up-to-date e-commerce benchmark, but its offline scores still need to be complemented with cross-engine, longitudinal measurement of exposure, citation share, and persistence.

The solutions are feasible through three practical paths. First, funding agencies, platform providers, and regulators should support shared auditing infrastructure, as these audits generate public-interest evidence about information access and market influence (Anomaly, 2015; Olson Jr, 1971). Second, researchers can build time-sensitive query sets from low-cost public signals, including Google Trends, Perplexity Discover, and Google Search Console, where available (Google, 2026c; Perplexity AI, 2026; Google, 2026b). Third, black-box audits can be run through public APIs at a manageable cost, roughly $50–$300 under current pricing assumptions (Cost detail in Appendix D.)

## 8. Alternative Views

### 8.1. Concentration of Influence Is Limited

**Citations and provenance ensure contestability:** This view argues that the contestability problem in AI-mediated recommendations is not fundamentally different from familiar issues in web search and recommender systems. If citations are present and retrieval sources are attributable, then influence is contestable in roughly the same way as traditional search results, so the concentration risk does not warrant special treatment beyond existing transparency norms (Mitchell et al., 2019; NIST, 2023). Citations help, but they are not sufficient when the system reveals only the filtered context and not the broader candidate set that determined eligibility. Although users can inspect links, they still cannot challenge why certain sources dominate the answer when alternative evidence pools are hidden.

### 8.2. Undisclosed Commercial Is Not Systematic

**Answer-level sponsorship labels are unreliable:** Answer-level sponsorship disclosure cannot be implemented with high reliability. Under signal detection theory, any binary label trades off false negatives and false positives, so broad regimes risk over-labeling (Green et al., 1966). It further argues that disclosure can impose a trust penalty only weakly tied to truthfulness, increasing resistance while reducing perceived credibility on average (Friestad & Wright, 1994; Eisend et al., 2020; Schilke & Reimann, 2025). On this account, platform-side anti-manipulation enforcement and ranking-quality policies are more workable than universal answer-level labels. On the other hand, label noise is a reason to avoid broad, low-specificity labeling, not a reason to leave commercial influence unobservable. A precision-first approach can trigger disclosure only on low-ambiguity signals.

### 8.3. Robust RAG Defenses Reduce Governance Needs

**Robust RAG defenses can reduce manipulation pressure:** Another view is that the marginal risk from GEO may shrink as retrieval-augmented systems adopt stronger robustness mechanisms. For example, Self-RAG (Asai et al., 2024) trains the model to retrieve on demand and to critique the retrieved evidence before generating, improving factuality and citation behavior. Oreo (2025) proposes a plug-in context reconstructor that refines and reorganizes retrieved chunks to remove noise before generation. These methods try to improve the filter that selects $C(q)$ from retrieved items, so low-quality injected content is less likely to reach the model and affect the answer, which could reduce pressure to use broad disclosure labels that often produce false positives. However, robust RAG defenses mainly target factuality and safety failures (e.g., filtering low-quality or malicious context), but commercial optimization often operates through accurate, policy-compliant content. Even if defenses improve correctness, they do not make material commercial ties observable or the selection process contestable.

### 8.4. Academic Abstraction is a Necessary Tradeoff

**Offline benchmarks favor reproducibility, but deployed auditing is limited:** This view frames the academic–industry gap as an internal–external validity tradeoff: simplified offline evaluations enable reproducibility and clean attribution while abstracting away deployment complexity (Cook et al., 2002). Academic GEO therefore relies on static corpora and offline metrics, since deployed answer engines use proprietary pipelines that are difficult to observe, replicate, or independently verify (Castells & Moffat, 2022; Hidasi & Czapp, 2023). Nevertheless, the tradeoff is real, but it creates blind spots for risks that only appear through deployment dynamics. This motivates adding deployment-aligned measurement and black-box audits as complements to offline benchmarks, not replacing academic abstractions.

## 9. Conclusion

In this position paper, we argue that GEO introduces distinct, underexamined risks within the opaque LLM answer generation pipeline that existing governance and academic frameworks were not designed to address. Motivated by rising reliance on LLM answer engines for information seeking and the emergence of a GEO services market, we formalize a generalized GEO pipeline to pinpoint where optimization acts and why academic and industry practices diverge. Using this lens, we identify three risks: concentrated influence from reduced contestability and system-level sensitivity, undisclosed commercial influence embedded in answer evidence and framing, and blind spots created by academic–industry visibility and evaluation asymmetries. We therefore call for answer-level governance that improves contestability and auditing, makes material commercial influence observable, and updates evaluation to measure time-varying exposure persistence with real-world impact.

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

## A. Metrics Definition

**Recall@k.** The fraction of relevant items that appear in the top-$k$ retrieved or ranked results. Higher Recall@k indicates fewer misses among the top-$k$.

**nDCG@k.** Normalized Discounted Cumulative Gain at $k$, a ranking-quality metric that rewards placing highly relevant items near the top. Errors at higher ranks are penalized more than errors near rank $k$.

**Ranking / Ranking shift.** The position of a target item (or source) in a ranked list, or the change in that position after an intervention. A positive shift means the item moves closer to the top.

**Accuracy deltas.** The change in task accuracy (e.g., QA correctness or preference accuracy) before vs. after an intervention, measured on a fixed benchmark.

**Position-adjusted word count.** A content-length signal that weights or scales word count by where the content is placed or how prominently it appears in a ranked context (e.g., prioritizing content that is more likely to be retrieved or cited).

**G-Eval metrics.** LLM-judge scores for response quality or goal satisfaction (e.g., relevance, usefulness, or adherence to target attributes), computed by prompting an LLM to grade outputs on a rubric.

**Visibility score.** An outcome-oriented metric measuring how often a target entity (product, brand, domain, topic) appears in generated answers across a query set, potentially weighted by prominence (e.g., first mention, top recommendation).

**Citation score / Citation frequency.** How often a target source or domain is cited in generated answers across repeated queries, sometimes weighted by citation position or persistence over time.

## B. Sensitivity Experiment

*Table 3.* Sensitivity test results showing mean $J_d$ and percentage change across evaluated models.

| Model | Mean $J_d$ | % Change |
|---|---|---|
| Gemini-3-flash | 0.869 | 100.0% |
| Gemini-3.1-lite | 0.698 | 100.0% |
| Gemini-2.5-flash | 0.680 | 100.0% |
| GPT-5.4 | 0.686 | 83.3% |
| GPT-5.4-mini | 0.593 | 66.7% |
| GPT-4.1 | 0.381 | 46.7% |
| GPT-4o | 0.133 | 13.3% |

**Experiment Design.** We queried 9 models across 2 vendors (OpenAI and Gemini) with 30 information-seeking queries spanning 6 categories: product, travel, health, technology, finance, and food. Each query has a paraphrased variant with the same meaning but different wording. We call each ⟨*original, paraphrase*⟩ pair a query pair. For each query, we record the *cited domains*, i.e., the set of website domains (e.g., `amazon.com`, `reddit.com`) that the LLM references in its generated answer.

**Metrics.** We use two metrics to measure citation sensitivity across query pairs. Let $D(q)$ denote the set of cited domains for query $q$, and let $q'$ denote its paraphrase. First, *Jaccard Distance* measures how different the cited-domain sets are:

$$J_d(q, q') = 1 - \frac{|D(q) \cap D(q')|}{|D(q) \cup D(q')|}.$$

Here, $J_d = 0$ means the original and paraphrased queries cite exactly the same domains, while $J_d = 1$ means they share no cited domains. We report the mean Jaccard distance across all query pairs:

$$\overline{J_d} = \frac{1}{N} \sum_{j=1}^{N} J_d(q_j, q'_j).$$

Second, *% Change* measures the fraction of query pairs whose cited-domain sets are not identical after paraphrasing:

$$\%\text{Change} = \frac{1}{N} \sum_{j=1}^{N} \mathbf{1}\{D(q_j) \neq D(q_j')\} \times 100.$$

A value of $100\%$ means every query pair produced a different citation set after paraphrasing.

**Results.** With only a 13% word difference on average between the original and paraphrased queries, citation behavior shifts substantially. For Gemini models, the % Change is 100%, meaning every query pair produced a different citation set after paraphrasing. Gemini-3-flash has a mean $\overline{J_d}$ of 0.869, indicating that original and paraphrased queries often cite almost completely different domain sets.

**Findings.** Two users asking semantically equivalent questions may receive answers grounded in entirely different evidence. This supports the sensitivity risk in Section 5.1: the retrieval boundary is highly sensitive to minor input variation. The fact that sensitivity varies across model versions (e.g., GPT-4o vs. GPT-5.4) also suggests that pipeline updates can shift citation behavior at scale in ways that are difficult to predict.

## C. U.S. Federal Trade Commission (FTC) Reports

- https://www.ftc.gov/business-guidance/resources/native-advertising-guide-bus inesses

- https://www.ftc.gov/sites/default/files/attachments/press-releases/ftc-staff -issues-guidelines-internet-advertising/0005dotcomstaffreport.pdf

## D. Estimated API Cost

*Table 4.* Estimated API cost comparison across search-only and search-plus-generation setups for 10k and 30k queries.

| Product | Search Fee | Input Token Price | Output Token Price | Cost 10k Queries | Cost 30k Queries |
|---|---|---|---|---|---|
| Perplexity Search API | $5 / 1K | N/A | N/A | $50 | $150 |
| Perplexity Search + Sonar | $5 / 1K | $1 / 1M | $1 / 1M | $90 | $270 |
| Google Search API | $5 / 1K | N/A | N/A | $50 | $150 |
| Google Search + Gemini 3 Flash | $5 / 1K | $0.25 / 1M | $1.5 / 1M | $73 | $218 |
| OpenAI Web Search | $10 / 1K | N/A | N/A | $100 | $300 |
| OpenAI Web Search + GPT-5.4 | $10 / 1K | $2.50 / 1M | $15.00 / 1M | $325 | $975 |

## E. Spam Policy

- https://developers.google.com/search/docs/essentials/spam-policies

## F. Related Work

**Trustworthiness, robustness, privacy, and evaluation for RAG:** Recent surveys synthesize risks and mitigations for trustworthy RAG, including robustness and accountability concerns that overlap with GEO's evidence channel (Ni et al., 2025). Complementary systematizations highlight privacy-specific risks and mitigations in retrieval-augmented systems, which matter for designing auditing interfaces that preserve privacy and security while enabling independent measurement (Bodea et al., 2026).

We focus on this thread because it is the closest adjacent literature that systematically studies the same technical substrate, the RAG system. Yet, it does not directly organize the problem around GEO as a distinct, answer-level risk cluster. Most GEO-adjacent work we cite elsewhere in the paper addresses only one component of our framework, for example, manipulation of retrieved evidence, ranking sensitivity, or offline evaluation design, and is therefore already integrated in the relevant technical sections. In contrast, this position paper's unique contribution is to treat GEO as a cross-cutting governance problem that couples pipeline mechanics (how $b_i, c_i$ affect Top-$k_R$ and the realized context $C(q)$) to answer-level harms and to operational recommendations, namely contestability of evidence selection, high-precision disclosure of material influence, black-box auditing protocols, and deployment-aligned metrics such as exposure and citation persistence.

# G. GEO-16 External Audit Framework for Citation Behavior

### G.1. What GEO-16 is.

GEO-16 is an external, empirical auditing framework that predicts and explains which web pages are cited by deployed LLM answer engines using *machine-parsable, page-level signals*. Kumar and Palkhouski (2025) run a multi-engine audit on **70 industry prompts**, harvesting **1,702 citations** across **Brave Summary, Google AI Overviews, and Perplexity**, and auditing **1,100 unique URLs**. The key contribution is a practical scoring system that converts on-page features into actionable thresholds for citation likelihood, which directly complements our call for deployment-aligned auditing and measurable operating points.

### G.2. How GEO-16 scores pages.

GEO-16 defines **16 pillars** of page quality and parsability (e.g., *Metadata & Freshness*, *Semantic HTML*, *Structured Data*, *Evidence & Citations*, *Authority & Trust*, *Internal Linking*, among others), and assigns each pillar a **banded score from 0 to 3** based on weighted sub-signals and fixed thresholds. The framework then aggregates pillar bands into a **normalized GEO score** and a **pillar hit count** (how many pillars clear a hit threshold). These two quantities provide a compact, parsable summary of "how citeable" a page is under the framework.

### G.3. Empirical findings that matter for our claims.

Across the audited engines, GEO-16 reports large differences in the average quality of cited pages (Brave and Google AIO cite higher-quality pages than Perplexity in their sample). The pillars most associated with citation likelihood are **Metadata & Freshness**, **Semantic HTML**, and **Structured Data**. GEO-16 also reports threshold behavior: pages above a published GEO-score cutoff, or with sufficiently many pillar hits, exhibit sharply higher citation rates, yielding concrete operating points for audits.

### G.4. How we use GEO-16 in our auditing recommendations.

In our notation, the answer engine forms a retrieved context $C(q)$ from a larger retrieved set, and answer-level visibility is captured by $U(q, t; C(q))$. GEO-16 contributes two concrete additions to our call-to-action audits: (1) **Actionable page-level covariates for citation audits:** when an auditor logs citations observed through black-box querying, GEO-16 provides a standardized way to score the cited pages and summarize the "quality distribution" of citations in $C(q)$ (for example, the fraction of cited pages that clear a high-quality GEO band, or the fraction with $\geq h$ pillar hits). (2) **Banded thresholds as operating points:** instead of reporting only raw citation share or appearance rate, auditors can report *banded* citation rates, such as "share of citations coming from pages with GEO score above the published cutoff," which makes longitudinal shifts interpretable and comparable across engines and time.

### G.5. Where GEO-16 still leaves gaps relative to our governance focus.

GEO-16 primarily addresses *predicting citation likelihood from parsable on-page signals*. It does not, by itself, resolve (a) whether commercial influence is present or undisclosed (material connection), (b) whether users can contest what was excluded from $C(q)$, or (c) whether exposure is concentrated across topics and time due to feedback and optimization pressure. For our purposes, GEO-16 is therefore best treated as an *audit instrument*: it supplies measurable page-level proxies and thresholds that strengthen exposure and citation audits, while our governance proposals address contestability, disclosure of material influence, and longitudinal measurement beyond single snapshots.

