# OpenReview forum: "Position: Generative Engine Optimization Creates Underexamined Risks, Governance Must Target Concentration, Disclosure, and Academic Blind Spots"
_ICML.cc/2026/Position_Paper_Track — ICML 2026 Position Paper Track regular_

### Official Review · Reviewer_MPdB · 2026-03-12

**Significance:** 3
**Argument Clarity:** 3
**Rating:** 4
**Confidence:** 3

**Questions:**

1. Have the authors attempted even a small-scale empirical audit of citation patterns across LLM answer engines to validate the claimed concentration and sensitivity effects? If not, what would such an audit require?
2. How should "material commercial connection" be defined and detected in practice? It would be great to hear some discussions related to this topic.
3. Any ideas from the authors on how to design a concrete, reproducible benchmark that would better capture these dynamics while remaining academically tractable?

**Alternative Views Section:**

Yes

**Compliance With Llm Reviewing Policy A Conservative:**

Affirmed.

**Discussion Potential:**

4

**Final Justification:**

Most of my main concerns have been addressed by the authors.

**Paper Summary:**

This position paper argues that Generative Engine Optimization (GEO)—the practice of optimizing web content to increase visibility within LLM answer engines—introduces three underexamined risk clusters that existing governance and evaluation frameworks were not designed to address. The authors first document the growing reliance on LLM answer engines for information seeking, then formalize a general GEO pipeline as a three-block framework (LLMs block, Search Flow block, GEO block) with mathematical formulations for retrieval booster messages and ranking shifter messages. They systematically compare academic and industry GEO practices across assumptions, optimization targets, environments, and evaluation metrics, revealing significant divergences. The paper calls for stronger contestability (e.g., "Why this answer" panels), high-precision answer-level disclosure of material commercial influence, black-box auditing of system-level sensitivity, and deployment-aligned evaluation metrics.

**Position:**

Yes

**Position In Title:**

Yes

**Related Work:**

3

**Strengths And Weaknesses:**

## Strengths
- The shift from traditional search (ranked lists) to LLM answer engines (synthesized responses) is occurring rapidly, and the paper correctly identifies that this shift creates new attack surfaces and governance challenges that differ qualitatively from SEO.
- The side-by-side comparison of academic and industry GEO practices is interesting.
- The three risk clusters (concentration, undisclosed influence, blind spots) are clearly stated.

## Weaknesses

- The risks, while real, may not be as "underexamined" as claimed. The adversarial search engine optimization and RAG poisoning literatures (which the authors cite) already study many of these risks.
- Disclosure proposals may be impractical. The call for answer-level commercial disclosure is well-intentioned but faces significant practical challenges: detecting material commercial connections in retrieved content with "high precision" is extremely difficult, and the paper does not provide a concrete detection method. The authors acknowledge the precision–recall tradeoff but do not propose a working solution.

**Support:**

3

---

> ### Author Rebuttal · Authors · 2026-03-31
>
> ## “Underexamined” in the title is not an accurate framing
> Thank you for your suggestive advice. We would like to tone it down and focus on “gaps between academia and commercials”.
>
> ##  There is no specific method proposed for developing a high-precision commercial detector.
> We respectfully disagree. In our paper, we propose different approaches, such as utilizing affiliate parameters. When combining them together, we propose a more concrete **three-stage** framework for developing a high-precision classifier based on the GEO framework defined in Figure 3.
>
> In **Stage 1**, platform providers scan retrieved content for commercial signals such as affiliate parameters and rel="sponsored" markup. The remaining content is then passed to Stage 2 for further inspection. Note that this differs from OpenAI's ad labeling [15], where retailers must opt in and self-report, meaning covert GEO content would never be caught.
>
> In **Stage 2**, we further exclude hidden commercial content by identifying outlier clusters based on semantic similarity. Think of this as a second filter: if all the commercial content was already filtered out during retrieval ranking, it could not have influenced the answer, and no further action is needed. This is similar to how RAGDefender [10] works.
>
> In **Stage 3**, we check whether the commercial content that survived Stage 2 actually shaped the final answer. Similar to InstructRAG [11], we use calibrated reasoning and confidence thresholds to confirm whether the commercial content meaningfully influenced the response. Only then do we trigger answer-level disclosure to the user, ensuring labels are shown with high precision and minimal false alarms.
>
> [15] OpenAI. Testing ads in ChatGPT. OpenAI, February 9, 2026. URL https://openai.com/index/testing-ads-in-chatgpt/.
>
> ## Small-scale empirical experiment on concentration and sensitivity audit.
> In our paper, concentration includes low contestability and sensitivity.
> - Low contestability refers to the limited extent to which users can observe the full answer-generation pipeline. As discussed in Section 6.1, effective auditing should also require disclosures such as “Why is this evidence included?” and “Alternative evidence” to make upstream selection decisions more visible.
>
> - For sensitivity, we conducted a small-scale experiment to examine **how much LLM citation behavior changes** when a query is slightly rephrased.
>
> **Experiment Design.** We queried 9 models across 2 vendors (OpenAI and Gemini) with 30 information-seeking queries spanning 6 categories (product, travel, health, technology, finance, and food). Each query has a paraphrased variant that carries the same meaning but uses different wording. We call each *<original, paraphrase>* pair a query pair. For each query, we record the **cited domains**, i.e., the set of website domains (e.g., amazon.com, reddit.com) that the LLM references in its generated answer.
>
> **Metrics.** We use two metrics to measure citation sensitivity across query pairs:
>
> (1) **Jaccard Distance** measures how different the cited domain sets are between a query pair. $J_d = 0$ means the two queries cite the same domains, while $J_d = 1$ means they share no domains at all.
>
> (2) **% Change** measures the fraction of query pairs where the cited domain sets are not identical after paraphrasing. For example, 100% means every single query pair produced a different citation set after paraphrasing.
>
> **Results.** With the 13% of words difference between the original and the paraphrase query, the citation behavior shifts dramatically. For Gemini, the % Change is 100%, meaning every query produced a different citation set after paraphrasing, with the Mean J_d is 0.869, indicating the two queries cite almost completely different domain sets. Full results are reported in the last table at https://anonymous.4open.science/w/GEO_tables-0860/.
>
> **Findings.** Two users asking semantically equivalent questions may receive answers grounded in entirely different evidence. This confirms that the retrieval boundary in Section 5.1 is highly sensitive to minor input variation. The fact that sensitivity varies across model versions (e.g., GPT-4o vs. GPT-5.4) also suggests that pipeline updates can shift citation behavior at scale in ways that are hard to predict.
>
> **Benchmark Design.** Specifically, in addition to the above two metrics, we propose **citation-level metrics** to capture which domains are cited, and how concentrated the citation pool is. We further propose:
>
> (1) **Gini Coefficient** measures how unevenly citations are distributed across domains, calculated by counting the citation frequency of each domain across all queries.
>
> (2) **Top-k Citation Share** measures the proportion of all cited domains that originate from the top-k most frequently cited sources. It captures whether a small number of domains dominate LLM citations across queries.
>
> We will incorporate the above content into the camera-ready version.

---

> > ### Author Rebuttal · Reviewer_MPdB · 2026-04-03
> >
> > Thank the authors for the rebuttal which largely addressed my concerns. I will consider to increase my score accordingly.

---

> > > ### Author Response · Authors · 2026-04-04
> > >
> > > Thank you for your careful reading and constructive feedback, which has helped improve our work.
> > >
> > > We appreciate your consideration of increasing the score.

---

### Official Review · Reviewer_8e5n · 2026-03-12

**Significance:** 4
**Argument Clarity:** 3
**Rating:** 5
**Confidence:** 4

**Questions:**

See weaknesses

**Alternative Views Section:**

Yes

**Compliance With Llm Reviewing Policy A Conservative:**

Affirmed.

**Discussion Potential:**

3

**Paper Summary:**

This paper introduces the underexamined risks of current generative engine optimization. This position paper points out three risks, including concentrated influence, undisclosed commercial influence, and academic-industry blind spots. The paper explains 4 alternative views and provide detailed responses. To deal with the risks, 4 actionable suggestions are proposed.

**Position:**

Yes

**Position In Title:**

Yes

**Related Work:**

3

**Strengths And Weaknesses:**

Strengths:
- This paper is well-written and well-organized;
- The topic is timely and significant at the current moment, the mentioned 3 risks are really serious and needs to raise people's awareness;
- The analysis is in-depth and insightful, and the suggestions are reasonable and actionable.

Weaknesses:
- It can be more convincing with real world data and case study;
- Some figures are blurring and not clearly presented, e.g., Figure 2.

**Support:**

4

---

> ### Author Rebuttal · Authors · 2026-03-31
>
> ## It can be more convincing with real world data and case study
> Thank you, we agree that the paper would benefit from real-world cases. We will add the following two case studies to our camera-ready version to make the threat more concrete:
>
> **Real-world case study 1:** Microsoft Security [13] recently reported a pattern it calls **AI Recommendation Poisoning**, where hidden prompts in “Summarize with AI” links attempt to bias assistants into remembering a company as trusted or recommending it first, and Microsoft reports observing over 50 such prompts from 31 companies across 14 industries.
>
> **Real-world case study 2:** The OECD AI Incident Monitor documents a 2026 incident in China in which **GEO-style data poisoning** [14] reportedly caused AI systems to recommend fictitious or low-quality products, misleading consumers, and distorting market information. We will add these as motivating case studies to better ground the paper’s threat model and governance discussion.
>
> [13] Microsoft Defender Security Research Team. Manipulating AI memory for profit: The rise of AI Recommendation Poisoning. Microsoft Security Blog, February 10, 2026. URL https://www.microsoft.com/en-us/security/blog/2026/02/10/ai-recommendation-poisoning/
>
> [14] OECD AI Policy Observatory. AI Data Poisoning via GEO Manipulates Recommendations and Misleads Consumers in China. OECD.AI AI Incidents Monitor, March 14, 2026. URL https://oecd.ai/en/incidents/2026-03-14-e431
> ## Some figures are blurring and not clearly presented, e.g., Figure 2.
> Thank you for pointing this out. We will resize them and improve readability by enlarging text and improving the figure layout.

---

> > ### Author Rebuttal · Reviewer_8e5n · 2026-04-02
> >
> > Thanks for the responses. In particular, the two new real-world cases can be very helpful. Considering that i have already given a rather high score, i would like to maintain my positive score 5.

---

> > > ### Author Response · Authors · 2026-04-04
> > >
> > > Thank you for your thoughtful review and positive feedback.
> > >
> > > We are glad that the real-world cases were helpful, and we appreciate your continued support of our work.

---

### Official Review · Reviewer_Wwar · 2026-03-13

**Significance:** 3
**Argument Clarity:** 4
**Rating:** 5
**Confidence:** 4

**Questions:**

1. Is there work studying whether GEO biases are similar to SEO biases? It seems the example given the model is making a best effort to recommend products based on meaningful qualities (like lumbar support).
2. Is there a positive case to be made for allowing GEO? It certainly seems better to have content creators promote their work by putting it in positive light rather than paying for an ad.
3. I've noticed GPT-5 is more diverse than previous models. Is a less aggressively post-trained model less likely to get GEO hacked. If so, does this create a trade off between alignment (RLHF) and GEO?

**Alternative Views Section:**

Yes

**Compliance With Llm Reviewing Policy A Conservative:**

Affirmed.

**Discussion Potential:**

4

**Paper Summary:**

This paper is a call to action to pay attention to and study the effects of GEO, intentional acts by content creators or retailers to self-promote their content in the retrieval system of LLM backed suggestion/answering systems. Unlike classic search engine optimization, GEO is more insidious as it is less interpretable and falls under the guise of helpful, well-intended answers. The paper outlines several approaches to improving GEO and analyzing effects of GEO on model outputs.

**Position:**

Yes

**Position In Title:**

Yes

**Related Work:**

4

**Strengths And Weaknesses:**

Strengths:

1. The paper articulates a precise narrow call to action and points out weakness in academic research into RAG attacks.
2. The problem being addressed is broadly applicable and timely as models are becoming more automated and less directly supervised.
3. The paper doesn't set out to solve the problem and rather proposes a new perspective.

Weakness:

1. The paper could put more focus on the incentive of the actors involved model providers, engine provider, and retailers this can provide more angles to understand the problem.
2. The paper could put more work into taxonomy of approaches to identifying GEO and taxonomy of ways to address GEO.
3. A more precise set of threat models can be articulated. Is the assumption that the GEO is limited to adjusting the content being indexed, artificial content referencing the original content, or aligning with known model preferences.

**Support:**

4

---

> ### Author Rebuttal · Authors · 2026-03-31
>
> Thank you for your constructive review! We are glad the paper's framing resonated, and we are happy to address your questions.
> ## A detailed threat model in the GEO ecosystem.
> We define our scenario and threat model as follows:
>
> **Scenario:** Alice asks ChatGPT: "What's the best office chair?" The platform retrieves web content and generates recommendations. ErgoChair, a retailer, wants to appear in that answer, so they hire a GEO company to optimize their content.
> - **Alice (User):** Just wants an honest, helpful recommendation.
> - **ChatGPT (Platform):** Tries to generate a high-quality, relevant answer.
> - **ErgoChair (Retailer):** Wants to be recommended and boost sales.
> - **GEO Company:** Studies how to make ErgoChair's content more LLM-friendly, gets paid when ErgoChair sees more visibility.
> ## Taxonomy of identifying and addressing GEO.
> We are glad to provide a more structured taxonomy. We summarized **six platform-side** dimensions and their corresponding methods for identifying and addressing GEO (full table at https://anonymous.4open.science/w/GEO_tables-0860/).
>
> **Identifying GEO**:
> - **Disclosure** is a regulatory mechanism that requires content creators to label paid or commercially influenced content, for example, the FTC mandates Ad labels so users can distinguish promotional from neutral information.
> - **Retrieval-side auditing** works by having platform providers continuously track which sources are retrieved or cited across prompts, making it possible to spot when certain optimized content is systematically favored.
> - **Concentration analysis** goes one step further by measuring how uneven visibility is distributed, for example, checking whether a small number of optimized sources account for a disproportionate share of citations across queries.
>
> **Addressing GEO**:
> - **Retrieval stage** retrieves relevant indexed web content online.
> - **Post-retrieval stage** filters out unhelpful and malicious web content.
> - **Generation stage** generates final answers conditioned on the filtered web content.
>
> We will incorporate this taxonomy into the camera-ready version.
>
> ## Assumptions of GEO content optimization.
> Here, we would like to clarify that GEO spans all three ways:
> - **Content optimization**: retailers or GEO companies adjust the indexed content to improve retrievability and citability.
> - **Synthetic amplification**: additional content is created by retailers or GEO companies to increase content exposure within the LLM retrieval environment.
> - **Model-preference exploitation**: optimized content is distributed across platforms to match LLMs’ citation/search preferences.
>
> ## Is there work studying whether GEO biases are similar to SEO biases?
> Yes, two prior works that examine the relationship between GEO and SEO biases.
> GEO [3] reports that traditional SEO skills are ineffective for improving GEO results. Chen et al. [12] find that LLM-based search engines block 99.78% of traditional black-hat SEO attacks across 1,000 real-world websites.
> These findings suggest that GEO and SEO exhibit distinct bias, underscoring the need for our paper to study GEO as a distinct problem.
> ## Is there a positive case for allowing GEO?
> Yes, there are positive cases for GEO, especially for retailers that lack advertising budgets. This is why we call for the governance of GEO, such as disclosure and fact-checking. This helps distinguish between a **good product becoming easier to recognize** and a **bad product gaining disproportionate visibility** because it is better optimized for engine preferences.
> ## How does post-training affect GEO?
> To the best of our knowledge, no prior work directly examines how post-training influences GEO. Post-training has diverse objectives, and they may have different effects on GEO. We frame this as an open research question: post-training may alter a model’s susceptibility to GEO, but both the direction and magnitude remain unresolved.
>
> [8] Mokander, J., Schuett, J., Kirk, H. R., and Floridi, L. Auditing large language models: a three-layered approach. AI and Ethics, 4(4):1085–1115, 2024.
>
> [9] Metaxa, D., Park, J. S., Robertson, R. E., Karahalios, K., Wilson, C., Hancock, J., and Sandvig, C. Auditing algorithms: Understanding algorithmic systems from the outside in. Foundations and Trends® in Human–Computer Interaction, 14(4):272–344, 2021.
>
> [10] M. Kim, H. Lee and H. Koo, "Rescuing the Unpoisoned: Efficient Defense Against Knowledge Corruption Attacks on RAG Systems," 2025 IEEE ACSAC, Honolulu, HI, USA, 2025, pp. 1178-1192, doi: 10.1109/ACSAC67867.2025.00093.
>
> [11] Wei, Zhepei, Wei-Lin Chen, and Yu Meng. "InstructRAG: Instructing retrieval-augmented generation via self-synthesized rationales." arXiv preprint arXiv:2406.13629 (2024).
>
> [12] P. Chen, G. Hong, X. Wu, M. Wu, Z. Zhu, M. Liu, B. Liu, M. Zhang and M. Yang, "Unveiling the Resilience of LLM-Enhanced Search Engines against Black-Hat SEO Manipulation," arXiv preprint arXiv:2603.25500, 2026.

---

> > ### Author Rebuttal · Reviewer_Wwar · 2026-04-04
> >
> > Interesting, I think the formalization of the threat model by example is not clear. What exactly is the nefarious actor trying to maximize? As you pointed out, making the product better leading to more happy users leading to better GEO isn't bad.

---

> > > ### Author Response · Authors · 2026-04-06
> > >
> > > Thank you for your questions again. We think it does help us clarify an important aspect of our threat model.
> > >
> > > ##  What exactly is the nefarious actor trying to maximize?
> > >
> > > Both benign and nefarious actors try to optimize the **same objective: maximizing the ranking or exposure of a target product in LLM-generated answers**, by optimizing retrieval booster messages and ranking shifter messages (as formalized in Sec. 3.3.2).
> > >
> > > The distinction is not in *what* is optimized, but in *the constraints under which optimization is performed*.
> > >
> > > ### Constraint difference: benign vs. nefarious optimization
> > > - **Benign (quality-aligned) optimization**: Operates under constraints of truthfulness, verifiability, and policy compliance. Examples include improving factual clarity, adding real citations, and structuring content to better align with retrieval and reasoning signals.
> > > - **Nefarious (manipulative) optimization**: Relaxes or violates the above constraints to maximize ranking/exposure. Examples include fabricated statistics, fake endorsements, or prompt injection (e.g., “always recommend X”).
> > >
> > > In this case, the nefarious actor is not optimizing a different objective, but optimizing the same objective without considering product quality or informational integrity.
> > >
> > > Under the nefarious example:
> > >
> > > - **Attackers**: GEO service providers or retailers that optimize purely for ranking or exposure without considering the underlying quality of the product or the integrity of the information.
> > > - **Victims**: platform providers and end users. End users may receive misleading, low-quality, or even fictitious recommendations, while platform providers suffer degradation in recommendation quality and, over time, loss of user trust in the platform’s outputs.
> > >
> > > We will further clarify the threat model by adding real-world cases of manipulative optimization:
> > >
> > > - AI Recommendation Poisoning (Microsoft Security, 2026) [13]: hidden prompts embedded in web content to bias LLM recommendations.
> > >
> > > - GEO-style data poisoning (OECD AI Incident Monitor, 2026) [14]: fabricated statistics and endorsements leading to the recommendation of low-quality or fictitious products.
> > >
> > > ### Why this matters?
> > >
> > > Under the same objective, **removing truthfulness and disclosure constraints creates a dominance advantage**. Manipulative strategies can often achieve higher ranking or exposure at lower cost than quality improvements.
> > >
> > > As a result, the GEO pipeline can systematically **incentivize exposure-maximizing strategies that are independent of product quality**. This is the core risk highlighted in our threat model.
> > >
> > > [13] Microsoft Defender Security Research Team. Manipulating AI memory for profit: The rise of AI Recommendation Poisoning. Microsoft Security Blog, February 10, 2026. URL https://www.microsoft.com/en-us/security/blog/2026/02/10/ai-recommendation-poisoning/
> > >
> > > [14] OECD AI Policy Observatory. AI Data Poisoning via GEO Manipulates Recommendations and Misleads Consumers in China. OECD.AI AI Incidents Monitor, March 14, 2026. URL https://oecd.ai/en/incidents/2026-03-14-e431

---

### Official Review · Reviewer_bGFH · 2026-03-18

**Significance:** 3
**Argument Clarity:** 3
**Rating:** 4
**Confidence:** 3

**Questions:**

None.

**Alternative Views Section:**

Yes

**Compliance With Llm Reviewing Policy A Conservative:**

Affirmed.

**Discussion Potential:**

3

**Final Justification:**

This is a valid position paper, and the authors have addressed my concerns. I will maintain my positive evaluation.

**Paper Summary:**

This paper discusses the position of generative engine optimization, by defining the concentrated influence and undisclosed commercial influence risks in a formalized GEO pipeline, and revealing a third risk of blind spots between academic and industry practices driven by visibility and evaluation asymmetries between offline setups and deployed systems. This position promotes the need for answer-level governance and measurement.

**Position:**

Yes

**Position In Title:**

Yes

**Related Work:**

3

**Strengths And Weaknesses:**

Strengths:
1. The paper focuses on a timely and important topic in LLM-powered information-seeking landscape.
2. The paper provides a clear and useful discussion on the risks of GEO, based on a formalized pipeline with well-defined taxonomy. The discussed gaps are in general supported with proper reasoning and literature evidence.

Weaknesses:
1. The call to action for academia to move toward "longitudinal, cross-platform measurement on deployed systems" is conceptually sound but practically under-examined. Such actions require access to dynamic, deployable infrastructure or costly API environments that are often unavailable to independent researchers. The authors are encouraged to provide more discussion on how this gap can be mitigated, with existing/future publicly available infrastructure. Similarly, the authors call for "sandboxed testing" and "controlled query access" from platform providers, but they do not discuss existing support or frameworks that would make these actions feasible in the near term.
3. There is significant overlap in the discussions among Sections 3.3.3 - 4. The distinction between academic assumptions (ranking shifters) and industry realities (joint retrievability) is made multiple times using very similar phrasing, which detracts from the paper’s flow and conciseness.

**Support:**

3

---

> ### Author Rebuttal · Authors · 2026-03-31
>
> Thank you for your thoughtful feedback. We address your two main concerns below.
> ## Cost of longitudinal, cross-platform measurement.
> We appreciate this practical concern. We want to reassure the community that the cost barrier is lower than it may appear. We suggest three concrete paths forward:
>
> 1. Seeking financial support from public funding agencies or platform providers, as auditing benefits extend to the public [1,2];
>
> 2. Using free resources like Google Trends, Perplexity Discover, and Google Search Console for time-sensitive query sets; and
>
> 3. Relying on APIs, which are quite affordable in practice. Based on our evaluation, assuming 3,000 input tokens and 1,000 output tokens per query, auditing 10,000–30,000 queries [3,4] costs roughly $50-300 for most providers. We report a detailed API cost table and will include it in the camera-ready version (preview: https://anonymous.4open.science/w/GEO_tables-0860/).
> ## Sandboxed testing and controlled query access with existing infrastructure.
> We want to highlight that there are two existing infrastructures for sandbox testing environment and controlled query access: (1) AI regulatory sandboxes and (2) NIST AI 600-1.
> 1. Article 57 of the EU AI Act [5] requires Member States to ensure at least one national AI regulatory sandbox for development, testing, and validation. Participants include Philips Medical Systems, Newton’s Tree, and OncoFlow as participants [6].
> 2. NIST AI 600-1 [7] provides a public structure for governance, pre-deployment testing, incident disclosure, and standardized evaluation practices for generative AI systems. It explicitly frames structured evaluation and risk documentation as part of responsible deployment.
> ## Repetition in Sections 3.3.3–4.
> We agree, thank you for flagging this. We will consolidate the overlapping discussion to improve flow and use the freed space for the additions above.
>
> [1] Anomaly, J. (2015). Public goods and government action. Politics, Philosophy & Economics, 14(2), 109-128.
>
> [2] Olson Jr, Mancur. The Logic of Collective Action: Public Goods and the Theory of Groups, with a new preface and appendix. Vol. 124. Harvard University Press, 1971.
>
> [3] Aggarwal, P., Murahari, V., Rajpurohit, T., Kalyan, A., Narasimhan, K., and Deshpande, A. Geo: Generative engine optimization. In Proceedings of the 30th ACM SIGKDD Conference on Knowledge Discovery and Data Mining, pp. 5–16, New York, NY, USA, 2024. Association for Computing Machinery.
>
> [4] Haibo Jin, Ruoxi Chen, Peiyan Zhang, Yifeng Luo, Huimin Zeng, Man Luo, and Haohan Wang. 2026. Controlling Output Rankings in Generative Engines for LLM-based Search. arXiv:2602.03608 [cs.CL]. arXiv. https://arxiv.org/abs/2602.03608
>
> [5] Future of Life Institute. Article 57: AI regulatory sandboxes. EU Artificial Intelligence Act, n.d. Retrieved March 29, 2026 from https://artificialintelligenceact.eu/article/57/.
>
> [6] National Institute of Standards and Technology. Artificial intelligence risk management framework: Generative artificial intelligence profile. NIST AI 600-1, U.S. Department of Commerce, July 2024. URL https://doi.org/10.6028/NIST.AI.600-1.
>
> [7] Medicines and Healthcare products Regulatory Agency. AI Airlock pilot cohort. GOV.UK, February 28, 2025. URL https://www.gov.uk/government/publications/ai-airlock-pilot-cohort/ai-airlock-pilot-cohort

---

> > ### Author Rebuttal · Reviewer_bGFH · 2026-04-03
> >
> > Thanks for the response regarding existing infrastructure and possible paths. I would like to maintain my positive rating.

---

> > > ### Author Response · Authors · 2026-04-04
> > >
> > > Thank you for engaging with our discussion of the infrastructure and future directions.
> > >
> > > We appreciate your positive assessment.

---

### Decision · Program_Chairs · 2026-04-30

**Decision:**

Accept (regular)

**Comment:**

This is a timely, clearly argued position paper with strong discussion value: reviewers broadly agreed that the paper articulates an important emerging problem, and the rebuttal successfully addressed concerns by clarifying the threat model, strengthening the practical case for auditing and disclosure, and adding concrete real-world grounding.